# Micro- and Small-Sized Enterprises' Sustainability-Oriented Innovation for COVID-19

**Weilun Huang [1], Hengbin Yin [1,*] , Seongjin Choi [2] and Mohsin Muhammad [3]**

[1]   School of Finance and Trade, Wenzhou Business College, Wenzhou 325035, China; huangwl@wzbc.edu.cn
[2]   School of Business, Hanyang University, Seoul 04763, Korea; seongjin@hanyang.ac.kr
[3]   College of International Finance and Trade, Zhejiang Yuexiu University of Foreign Languages,
     Shaoxing 312000, China; mohsingrw@hotmail.com
*   Correspondence: 20190251@wzbc.edu.cn; Tel.: +86-158-5855-7931

**Abstract:** The economic impact of a public emergency, such as the COVID-19 pandemic, is often reduced by micro and small businesses (MSEs) undertaking sustainability-oriented innovation for public emergencies (SOIPE), which includes production and service innovation, information innovation, marketing innovation, and labor innovation. The originality of this study lies in its prediction and evaluation of COVID-19's challenges and SOIPE's requirements to have a keen observation and discovery ability. In this paper, we combined nominal group technique, fuzzy analytical hierarchy process, least squares, and a case study to investigate governance, economic, financial, sociocultural, and environmental sustainability and demonstrate the MSEs' sustainability evaluation model. In a qualitative study and literature review, MSEs were found to use SOIPE in a variety of ways. Some studies focused on marketing innovation, while others were hampered by their limited understanding. From both a theoretical and empirical perspective, this study suggests that MSEs should identify their optimal SOIPE based on the impact and volatility of a public emergency. In addition, this study presents an assessment of the impact and environmental volatility of a public emergency, as well as MSEs' SOIPE, which is more helpful for enterprises. Finally, this study creatively introduces the SOIPE of MSEs, which has important policy ramifications.

**Keywords:** micro- and small-sized enterprises; sustainability-oriented innovation for public emergency; public emergency; sustainability





## 1. Introduction

The COVID-19 pandemic has caused short-term setbacks for many small- and medium-sized enterprises (MSEs) in many countries. MSEs are therefore considering their sustainability post-COVID-19. Therefore, the issue of sustainability should be an essential issue for modern businesses, measured by their corporate social responsibility (CSR) [1]. However, MSEs may care more about economic CSR than social and environmental CSR. The most direct impact of the COVID-19 pandemic is a diminished level of customer demand, which concerns most MSEs [2]. As a result of the economic recession aggravated by the COVID-19 pandemic, MSEs have been negatively impacted [3–5]. It has been argued that the economic and financial effects of the COVID-19 pandemic should be described as a negative impact and perceived risk to the environment. MSEs make up 90% of the total number of enterprises in many countries and regions. The economic development mode of MSEs is rapidly changing as they continue to grow on a global scale, as is their employment absorption capacity. Their role has been to propel regional growth and maintain social stability. Apart from providing jobs, MSEs are responsible for most technological and managerial innovations and contribute to the development of the economy [6–8].

Existing studies suggest that public emergencies' sustainability evaluation criteria should include governance sustainability (GS), economic sustainability (ES), financial sustainability (FS), and sociocultural and environmental sustainability (SS) [9,10]. Furthermore,

during the COVID-19 pandemic, some MSEs used sustainability-oriented innovation for public emergencies (SOIPE), which comprises production and service innovation ($SOIPE_{PS}$), information innovation ($SOIPE_I$), marketing innovation ($SOIPE_M$), and labor innovation ($SOIPE_L$). In addition, MSEs incorporate product, process, and marketing innovations to improve their marketing, financial, and production performances [3,11]. Therefore, the MSEs should be able to respond efficiently and effectively to public emergencies through SOIPE, which would lead to improved performance, competitiveness, and risk management [3,12]. In most studies, MSE innovations are not distinguished by type; however, with limited resources, the costs and benefits of a variety of types of innovations should differ, especially in relation to budgetary constraints. Innovation is said to include new technologies, products, processes, services, and management concepts [6].

Several studies have discussed the effects of $SOIPE_{PS}$, $SOIPE_I$, $SOIPE_M$, and $SOIPE_L$ on MSE performance through the key dimensions of GS, ES, FS, and SS. However, fewer studies have focused on the mediating effects of $SOIPE_{PS}$, $SOIPE_I$, $SOIPE_M$, and $SOIPE_L$, for MSEs, which may improve the causal relationships between public emergencies (e.g., COVID-19 pandemic) and MSE performance in GS, ES, FS, and SS. In addition, fewer studies have examined why MSEs need to innovate to cope with the impact of the environmental volatility of a public emergency, and none have examined how MSEs should choose the best SOIPE to respond to the impact. A company's product and service innovation can, on the other hand, transform its growth, profitability, and competitiveness [13,14]. In the published literature, production and service innovation play a vital role in responding to volatility in the market, supply chain, and socioeconomic environment; however, such a response might not be appropriate for MSEs [13,15].

In light of the above discussion, this article focuses on the following research questions: (1) the impact and environmental volatility MSEs perceived during the COVID-19 pandemic, as measured by GS, ES, FS, and SS; (2) impact of $SOIPE_{PS}$, $SOIPE_I$, $SOIPE_M$, and $SOIPE_L$ on MSE performance, as measured by GS, ES, FS, and SS; and (3) MSEs' theories and procedures for choosing the optimal SOIPE for the impact and environmental volatility of a public emergency. Thus, the purpose of this study is to connect theories of innovation in MSEs with economic impact theory related to public emergencies. MSEs are provided with recommendations on how to select the most suitable innovation when responding to the economic impacts of public emergencies. Research gaps on public emergencies, including the SOIPE, are also addressed in the present study. Through this study, economies can develop sustainably in many ways. Governing bodies and MSEs must first explore how MSEs optimize SOIPE for the impact and volatility of a public emergency; then, they can determine the best policies to help MSEs survive a public emergency. Second, it is also important for scholars to explore the theories of MSEs in order to select the best SOIPE for the impact and environmental volatility of a public emergency, allowing them to find the best connection between the theories.

Hence, the overall aim of this study is to identify what is the best SOIPE for MSE perceptions of COVID-19 effects and environmental volatility, which is evaluated and selected from $SOIPE_{PS}$, $SOIPE_I$, $SOIPE_M$, and $SOIPE_L$. In this study, we first set the key dimensions of GS, ES, FS, and SS, then examine the causal relationships between a public emergency (i.e., the COVID-19 pandemic) and MSE performance. Secondly, we examine the mediators of SOIPE in the causal relationship between public emergencies and MSE performance. Third, we evaluate the mediating effects of MSEs' $SOIPE_{PS}$, $SOIPE_I$, $SOIPE_M$, and $SOIPE_L$ in the key dimensions of GS, ES, FS, and SS. The remainder of this paper is organized as follows. Section 2 discusses the literature on MSEs' perceived environmental volatility and negative impacts of COVID-19 and MSEs' $SOIPE_{PS}$, $SOIPE_I$, $SOIPE_M$, or $SOIPE_L$. The comprehensive evaluation model and procedure for MSEs' sustainability evaluation are set in Section 3 based on a literature review and expert survey. In Section 4, we analyze the results of the comprehensive evaluation model using a case study of Chinese MSEs and summarize the theoretical and management implications, as well as the limitations of this study. The conclusion and future research directions are outlined in Section 5.

## 2. Literature

The research design of this study is to match MSEs' perceived challenges of COVID-19 with MSEs' SOIPE$_{PS}$, SOIPE$_I$, SOIPE$_M$, SOIPE$_L$, so the hypothesis of this study should be MSEs' SOIPE$_{PS}$, SOIPE$_I$, SOIPE$_M$, or SOIPE$_L$ is the best SOIPE to cope with MSEs' perceived effects and environmental volatility from the COVID-19 pandemic. By considering emergency-impacted MSE capabilities, we developed SOIPE for MSEs. To accomplish this, MSEs can harness the perceived environmental volatility and negative impacts of a public emergency to develop and implement an effective SOIPE. MSEs' SOIPE is also the first study to include it as a mediator between the perceived negative impact of a public emergency and the environment's volatility and their ability to perform. MSE performance and perceived negative effects of a public emergency can be better understood by this study, and SOIPE can be explored as a mediating factor. MSEs' pro-emergency practices are also influenced by a growing number of studies examining the perceived negative impacts and environmental volatility of public emergencies. Through the mediating mechanism of SOIPE, we show causal relationships between perceived negative impacts, environmental volatility, and MSE performance. Emergency situations can adversely impact MSE performance, which leads to MSEs adopting sustainability-oriented innovations to alleviate those causal relationships. MSE performance and perceived environmental volatility are only rarely correlated [16–18]. To enhance MSE performance and address perceived environmental volatility, a selection mechanism could choose the optimal SOIPE for incremental improvements. Cao et al. (2022) [6] reported that return on assets is a measure of the performance of MSEs.

COVID-19 has primarily impacted MSEs in relation to economic activities, unemployment, trade, supply chains, businesses, and personal networks, as well as cross-border transfers of knowledge, technology capital, ideas, and people. Moreover, long-term effects include secular stagnation caused by precautionary saving or rebuilding depleted wealth, as well as the joint effect of the above negative effects across different regions. According to Achi et al. [3], green innovation processes balance and adjust perceived changes in the environment. In their report, Twahirwa et al. [19] noted that MSEs faced challenges due to the COVID-19 pandemic primarily related to client bills, investment ability, input accessibility, and consumer accessibility. According to the International Trade Centre [10], COVID-19 has shutdown impacts, supply chain disruptions, and demand depressions on MSEs.

Price, demand, and supply chain volatility are increasing more than ever before as a result of this public emergency. Furthermore, customer preferences, technology, and investor risk aversion in domestic and foreign capital markets play a role. It makes it difficult for MSEs to predict profit trends [3,20]. MSEs' external environment volatility can include unpredictable market demand, unstable production volume, and difficulty monitoring market prices, according to Achi et al. [3]. According to Maldonado-Romo and Aldape-Perez [21], the COVID-19 pandemic could reduce MSE profitability.

Among the perceived advantages of MSEs are production cost-effectiveness, recycling and reusing materials and parts, and the environment-safe disposal of hazardous substances and waste [3,22]. Compared with other SOIPE types, MSEs perceive the disadvantages of SOIPE$_{PS}$ to be higher capitalized costs, longer period, and more uncertainty of market acceptance. In Mexico, Donovan et al. [23] found that most local maize seed MSEs focus on their limited marketing investments, production infrastructure, and business management. Păunescu and Mátyus [24] found that Romanian MSEs coped with the disruption caused by the COVID-19 pandemic through SOIPE$_{PS}$, which includes production innovation and demand support for customers and communities.

The advantages of SOIPE$_I$ perceived by MSEs are that various resources and information can be shared and used quickly, and production and marketing can be improved effectively. MSEs can improve their allocation efficiency for labor, knowledge, management, capital, and technology by improving information efficiency. MSEs could improve allocation distortion by participating in the value distribution of economic chains. Compared

with other SOIPE types, MSEs perceived disadvantages of SOIPE$_I$, such as higher costs for information securities, more information personnel, and more training for information and communication skills. Da Costa et al. [25] found that MSEs could optimize process routine performativity through human cognition, which could promote by SOIPE$_I$. Antypenko et al. [26] reported that information support from SOIPE$_I$ could help MSE development, considering MSEs' operating problems in relation to organization, communication, technology, markets, and information. Dierckx and Stroeken [27] posited that cooperation and networking should be the advantages of MSEs' SOIPE$_I$. Păunescu and Mátyus [24] found that MSEs' SOIPE$_I$ might promote their performance, the targets of which include supply chain stabilization and broadening access to relevant information.

Advantages MSEs perceive for SOIPE$_M$ are improved efficiency and reduced costs. MSEs could make full use of the Internet to shorten their value chains or use online social media to establish consumer communities and reverse customize products. MSEs perceive the disadvantages of SOIPE$_M$ to be its higher costs for transaction securities, higher trust and search costs for consumers, and higher transaction costs of laws and regulations. Jeong and Chung, Aksoy, Gupta et al., and Naidoo [28–31] argued that SOIPE$_M$ would be improvements in the marketing mix, including products, placement, promotion, or pricing. An MSE's ability to absorb new processes and technologies should be determined by its market, processes, and marketing innovations, according to Taques et al. [32].

For SOIPE$_L$ MSE perceives advantages in its flexible labor allocation, accurate labor cost-control, quick labor efficiency improvement, and effective employment risk-reduction. On the labor side, SOIPE$_L$ could help MSEs achieve maximum labor value and diversified development. However, MSEs perceive disadvantages of SOIPE$_L$ such as higher committed labor costs, labor adjustment costs, and labor adaptation costs. Bartlett and Morse [33] found that most MSEs have used the labor flexibility from SOIPE$_L$ to survive during the COVID-19 pandemic; however, there are higher closure risks from committed labor costs. Păunescu and Mátyus [24] found that SOIPE$_L$ might promote MSE performance, including operational management efficiency, worker protection, workplace safety, and working conditions.

## 3. Methodology of Micro- and Small-Sized Enterprises' Sustainability Evaluation

It should be an issue for MSEs to find the best SOIPE to overcome challenges or seize opportunities and then increase the success rate of sustainable development for MSEs. As part of their sustainable development, MSEs are going to encounter many events that can cause abrupt and random changes in environmental conditions (such as COVID-19), which they will then have to cope with. As a result, MSEs, on the whole, would use the SOIPE to improve their responses to challenges and opportunities, for SOIPE might reduce MSEs' costs or increase their benefits in the long run or in the short run; on the other hand, SOIPE might change MSEs' organizations, processes, and labors' desire and passion for enabling sustainable development of MSEs.

In order to be able to predict challenges (or opportunities) and SOIPE, MSEs and governments must have a good comprehensive evaluation model with very keen observational abilities. A correctly predicted challenge (or opportunity) and SOIPE are of great importance during decision-making for MSEs, and if the prediction is not accurate, the MSE may face disaster. Therefore, academia, practitioners, and MSEs should benefit from the MSEs' sustainability evaluation model and procedure.

### 3.1. Analysis Structure of the Methodology

The MSEs' sustainability evaluation model and procedure were developed based on the results of a literature review and a survey of experts. There is a strong correlation between dimensions and criteria in this approach, and it can provide reliable expert opinions at relatively low costs [16,17].

Four steps make up the comprehensive evaluation model for MSE sustainability. First, we gathered a list of sustainability evaluation dimensions from various scholars.

Management, economy, finance, science, environmental volatility, and SOIPE perceptions were studied. Second, we used NGT to produce the final criteria for the MSE sustainability evaluation. Its advantages include time-saving, applicability to nominal and interacting groups, and equality among members [18].

For the third step, we prepared a FAHP questionnaire and conducted six questionnaire surveys with MSE employee supervisors, combined with the triangular fuzzy number to construct the evaluation dimensions and criteria of MSE sustainability [16,17]. The topics of six questionnaire surveys were to evaluate MSEs' perceived impacts on COVID-19, perceived environmental volatility from the COVID-19 pandemic, and perceived influence from $SOIPE_{PS}$, $SOIPE_I$, $SOIPE_M$, $SOIPE_L$ based on the criteria of the MSE sustainability evaluation. Then, we used Analytic Hierarchy Process Expert Choice 2000 software to find six importance evaluations ($I_{im}$, $I_{vo}$, $I_{S_{PS}}$, $I_{S_I}$, $I_{S_M}$, $I_{S_L}$) of the final criteria of the MSE sustainability evaluation from six questionnaire surveys, which are MSEs' perceived impacts and perceived environmental volatilities to COVID-19, and perceived influence from $SOIPE_{PS}$, $SOIPE_I$, $SOIPE_M$, $SOIPE_L$. For simplicity, $\sum_i I_{im} = \sum_i I_{vo} = \sum_i I_{S_{PS}} = \sum_i I_{S_I} = \sum_i I_{S_M} = \sum_i I_{S_L} = 1000$ were assumed.

In the fourth step, we used least-squares methods to find MSEs' best SOIPE for perceived effects and environmental volatility from the COVID-19 pandemic. The evaluation criteria of MSEs' best SOIPE is the minimum of the difference square on $I_{S_{PS}}$, $I_{S_I}$, $I_{S_M}$, or $I_{S_L}$ ($Q_{S_{PS}}$, $Q_{S_I}$, $Q_{S_M}$, or $Q_{S_L}$), which is compared with $R_{im}$ and $R_{vo}$. The equations are as follows:

$$\text{Lookup } \left\{ SOIPE_{PS}, \ SOIPE_I, \ SOIPE_M, \ SOIPE_L \left| \text{Min} \left( Q_{S_{PS}}, \ Q_{S_I}, \ Q_{S_M}, \ Q_{S_L} \right) \right. \right\} \tag{1}$$

$$\text{S.t. } Q_{S_{PS}} = \sum_i \left( I_{S_{PS},i} - I_{im,i} \right)^2 + \sum_i \left( I_{S_{PS},i} - I_{vo,i} \right)^2, \tag{2}$$

$$Q_{S_I} = \sum_i \left( I_{S_I,i} - I_{im,i} \right)^2 + \sum_i \left( I_{S_I,i} - I_{vo,i} \right)^2, \tag{3}$$

$$Q_{S_M} = \sum_i \left( I_{S_M,i} - I_{im,i} \right)^2 + \sum_i \left( I_{S_M,i} - I_{vo,i} \right)^2, \tag{4}$$

$$Q_{S_L} = \sum_i \left( I_{S_L,i} - I_{im,i} \right)^2 + \sum_i \left( I_{S_L,i} - I_{vo,i} \right)^2, \tag{5}$$

where *i* is the influencing ranks in the criteria for the MSE sustainability evaluation dimensions (Figure 1).

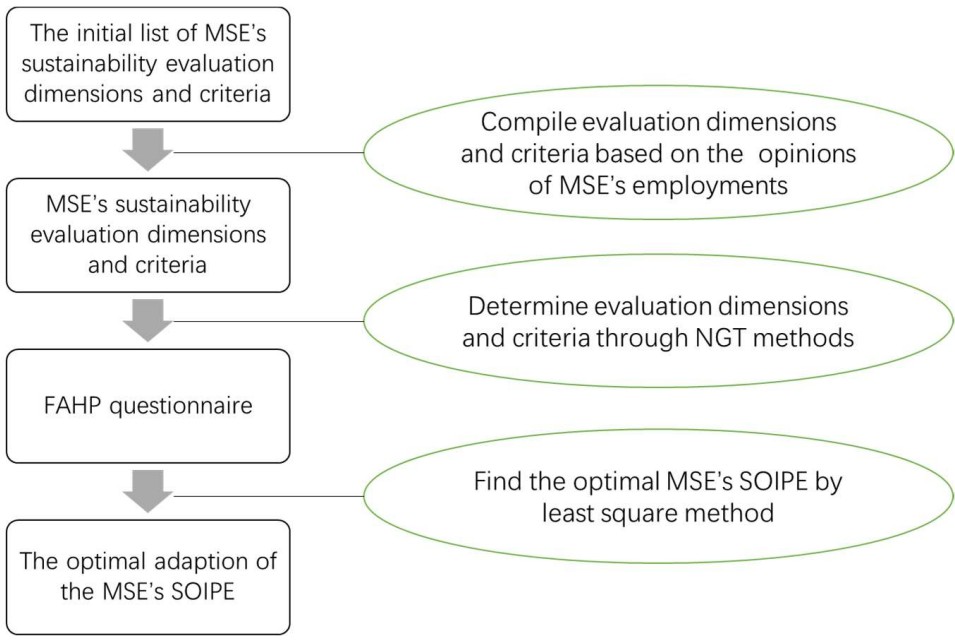

**Figure 1.** The comprehensive evaluation model and procedure for sustainable island tourism.

### 3.2. Initial List of Micro- and Small- Sized Enterprises' Sustainability Evaluation Criteria

According to this study, the literature review and expert survey established an initial list of MSE sustainability evaluation criteria, based on MSEs' perception of the COVID-19 pandemic and MSEs' perception of SOIPE advantages and disadvantages.

From the above discussion in Section 2 and NGT results, this study set the dimensions of MSE sustainability evaluation criteria compress of GS, ES, FS, and SS. The advantages and disadvantages of SOIPE perceived by MSEs include governance power, governance risks, economic growth, economic risks, financial growth, financial risks, sociocultural risks, and environmental risks, which should be the influencing factors of the perceived impact of the COVID-19 pandemic in the GS, ES, FS, and SS dimensions.

### 3.3. Micro- and Small- Sized Enterprises' Sustainability Evaluation Criteria

The final MSE sustainability evaluation criteria are listed in Tables 1 and 2. As Table 1 shows, the subdimensions of GS in this study are MSE innovation governance organizations ($GS_1$) which would affect its governance power, and avoidance of governance risks ($GS_2$). The indicators of $GS_1$, which are official, semiofficial, and unofficial innovation governance organizations ($GS_{11}$, $GS_{12}$, $GS_{13}$), aim to analyze their costs and benefits on MSEs' impacts of COVID-19 and the advantages and disadvantages of SOIPE perceived by MSEs. The indicators of $GS_2$, which are policies, laws, and regulations ($GS_{21}$); budgets, taxes, and subsidies ($GS_{22}$); and governance structure and efficiency ($GS_{23}$), aim to evaluate the avoidance of governance risks on MSEs' impacts of COVID-19 and the advantages and disadvantages MSEs perceive for SOIPE [34–36].

**Table 1.** Governance and economy and finance dimensions of MSE sustainability evaluation criteria.

| Dimensions | Subdimensions | Criteria |
|---|---|---|
| Governance Sustainability (GS) | MSE Innovation Governance Organizations ($GS_1$) | Official Innovation Governance Organization ($GS_{11}$) |
| | | Semiofficial Innovation Governance Organization ($GS_{12}$) |
| | | Unofficial Innovation Governance Organization ($GS_{13}$) |
| | Avoidance of Governance Risks ($GS_2$) | Policies, Laws, and Regulations ($GS_{21}$) |
| | | Budgets, Taxes, and Subsidies ($GS_{22}$) |
| | | Governance Structure and Efficiency ($GS_{23}$) |
| Economic Sustainability (ES) | Economic Growth ($ES_1$) | Long-Term Economic Growth ($ES_{11}$) |
| | | Short-Term Economic Growth ($ES_{12}$) |
| | | Economic Efficiency ($ES_{13}$) |
| | Avoidance of Economic Risks ($ES_2$) | Interruption of Supply Chain ($ES_{21}$) |
| | | Interruption of Demand Chain ($ES_{22}$) |
| | | Industrial Competitiveness and Product and Service Quality ($ES_{23}$) |

The subdimensions of ES in this study are economic growth ($ES_1$) and avoidance of economic risks ($ES_2$). The indicators of $ES_1$, which are long- and short-term economic growth ($ES_{11}$, $ES_{12}$) and economic efficiency ($ES_{13}$), aim to analyze their economic influence from MSEs' impacts of COVID-19 and the advantages and disadvantages MSEs perceive for SOIPE. The indicators of $ES_2$, which are the interruption of the supply chain and demand chain ($ES_{21}$, $ES_{22}$) and industrial competitiveness and product and service quality ($ES_{23}$),

aim to evaluate their avoidance of economic risks on MSEs' impacts of COVID-19 and the advantages and disadvantages MSEs perceive for SOIPE [37–39].

**Table 2.** Financial, sociocultural, and environmental dimensions of MSE sustainability evaluation criteria.

| Dimensions | Subdimensions | Criteria |
|---|---|---|
| Financial Sustainability (FS) | Financial Growth ($FS_1$) | Profitability ($FS_{11}$) |
| | | Cost-Down ($FS_{12}$) |
| | | Financial Efficiency ($FS_{13}$) |
| | Avoidance of Financial Risks ($FS_2$) | Cash Flow ($FS_{21}$) |
| | | Seasonality ($FS_{22}$) |
| | | Diversity ($FS_{23}$) |
| Sociocultural and Environmental Sustainability (SS) | Avoidance of Sociocultural Risks ($SS_1$) | Underemployment ($SS_{11}$) |
| | | Local Antagonism ($SS_{12}$) |
| | | Local Cultural Assimilation and Exploitation ($SS_{13}$) |
| | Avoidance of Environmental Risks ($SS_2$) | Pollution-Avoidances of Air, Water, and Land ($SS_{21}$) |
| | | Recycling Waste and Sewage ($SS_{22}$) |
| | | Reduced Use of Natural Resources and Biodiversity ($SS_{23}$) |

The subdimensions of FS in this study are financial growth ($FS_1$) and avoidance of financial risks ($FS_2$). The indicators of $FS_1$, which are MSE profitability ($FS_{11}$), cost-down ($FS_{12}$), and financial efficiency ($FS_{13}$), aim to analyze their financial discrepancy on MSEs' impacts of COVID-19 and the advantages and disadvantages of SOIPE perceived by MSEs. The indicators of $FS_2$, which are MSEs' cash flow ($FS_{21}$), seasonality ($FS_{22}$), and diversity ($FS_{23}$), aim to evaluate the avoidance of financial risks in relation to the impact of COVID-19 on MSEs and the advantages and disadvantages of SOIPE perceived by MSEs [40,41].

The subdimensions of SS in this study are the avoidance of sociocultural ($SS_1$) and environmental risks ($SS_2$). The indicators of $SS_1$, which are MSE underemployment ($SS_{11}$), local antagonism ($SS_{12}$), and local cultural assimilation and exploitation ($SS_{13}$), aim to analyze their sociocultural risks' avoidance on MSEs' impacts of COVID-19 and the advantages and disadvantages of SOIPE perceived by MSEs. The indicators of $SS_2$, which are avoiding air, water, and land pollution ($SS_{21}$); recycling waste and sewage ($SS_{22}$); and reducing natural resource use and biodiversity ($SS_{23}$), aim to evaluate the environmental risks' avoidance on MSEs' impacts of COVID-19 and the advantages and disadvantages of SOIPE perceived by MSEs [42–45].

## 4. Case Study of Chinese Micro- and Small-Sized Enterprise

### 4.1. Background of Chinese Micro- and Small- Sized Enterprise

MSEs play a key role in China's economy. Approximately 90% of enterprises and 70% of workers in China were MSEs by 2021. Over 60% of China's GDP was derived from them, and over half of its taxes were paid by them. There are certain criteria that Chinese MSEs must meet: (1) total assets below RMB 30 million, or RMB 10 million for industrial MSEs and nonindustrial MSEs, (2) no more than 100 or 80 employees, and (3) annual taxable revenue not exceeding RMB 300 thousand. The registration and cancellation ratio in China has declined as a result of COVID-19, although registrations continue to exceed cancellations. The number of MSE registrations has declined significantly, and more than 10% have declared bankruptcy. Registrations for business services, real estate, and information transmission are lower than cancellations.

In China, MSEs are experiencing a decline in prosperity in 2022, and their operations are being pressured. As a result, various operation indicators of MSEs have deteriorated in China, such as the purchasing manager's index and the small and medium enterprises development index. As a result of the soaring sea freight and the rigid rise in labor costs, MSEs' production and operation costs have continued to rise, as have the prices of international bulk commodities and the cost of energy. Production and sales are facing great uncertainty, which may explain why MSEs have difficulty getting orders. A large number of MSEs will not take large orders and will not expand production capacity. Second, small and micro enterprises feel the pressure of order transfer as the international industrial supply chain slowly recovers. Third, MSEs have slowed down the collection of sales loans, increased receivables, occupied current assets more, and tightened the capital chain.

In China, MSEs have contributed over half of all patents but do not invest enough in research and development. Data platforms and deep data mining are among MSEs' information management features. Information management systems are connected through design, procurement, production, manufacturing, finance, marketing, operations, and management. There are more MSEs registered in south China, where the economy is more marketized than in north China, where the economy is less marketized. MSEs have increased their use of intelligent manufacturing and flexible production to cope with the fluctuating labor needs caused by COVID-19.

### 4.2. Step 3: FAHP Questionnaire Survey

This study reviewed relevant literature and examined the background of Chinese MSEs, which were the references for the FAHP questionnaire on sustainable island tourism. The FAHP questionnaire surveys were submitted by 16 Chinese business owners or representatives who have been in operation for at least three years. The COVID-19 pandemic killed many MSEs; therefore, if an MSE has operated for more than three years, this should indicate that the MSE has better SOIPE. 96 questionnaires on six topics were collected, and expert evaluation opinions were compiled. Surveys were conducted from 1 February to 26 March 2022. To ensure data reliability and accuracy, questionnaires were administered during face-to-face interviews.

In this study, respondents were selected by stratified random sampling by industry of MSEs and by the length of time they have been working for the organization. Respondents work at MSEs mainly in wholesale and retail, production and processing, service, construction, and transportation. Approximately 71% of customers are in wholesale and retail, production and processing, and service.

Descriptive statistics for the questionnaire respondents are presented in Table 3. One-fourth were managers/senior workers, and three-fourths were associate workers. One-fourth of the respondents had a master's degree or above, while three-fourths had a bachelor's degree or below. Respondents had been employed at their MSE for an average of 5.25 years (standard deviation: 0.25). Of the respondents, 37.50% worked for MSEs located in a first-tier city (Beijing, Shanghai, Guangzhou, and Shenzhen), while 62.50% worked for MSEs located in a second- or third-tier city.

**Table 3.** Frequency analysis.

|  |  | Frequency |
|---|---|---|
| Position | Managers/senior workers | 12 |
|  | Associate workers | 4 |
| Education | Master's degree or above | 4 |
|  | Bachelor's degree or below | 12 |
| Duration working for their MSE | Between 3 and 5 years | 6 |
|  | More than 5 years | 10 |
| MSE Area | First-tier cities (Beijing, Shanghai, Guangzhou, and Shenzhen) | 6 |
|  | Second- or third-tier city | 10 |

For a brief introduction to FAHP, a fuzzy set is characterized by a membership function, and general terms will be used to capture a range of numerical values [46]. A fuzzy number M is described as a fuzzy subset of the real line x, with a utility function with uncertainty, $\mu_M(x)$. This membership function is defined in a universe of discourse of [0, 1]. Thus, a fuzzy triangular number (Figure 2) can be defined as a triplet $(a, b, c)$, where $a \leq b \leq c$. The parameters $(a, b, c)$ are the least possible, most possible, and largest possible values. The membership function is defined as:

$$\mu_M(x) = \begin{cases} x - a/b - a, \ x \in [a, b] \\ c - x/c - b, \ x \in [b, c] \\ \quad\quad 0, \ otherwise \end{cases} \tag{6}$$

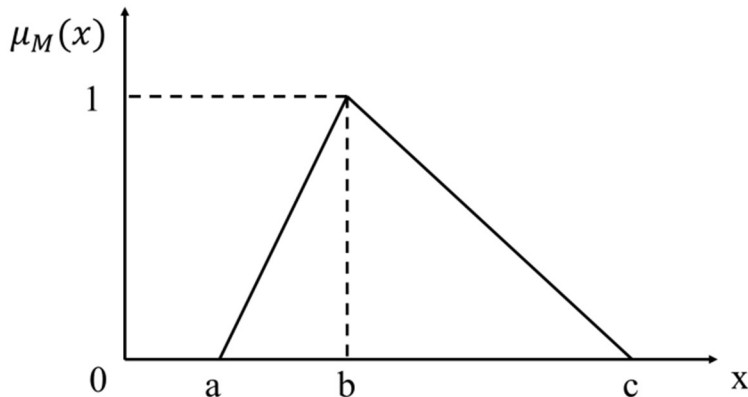

**Figure 2.** The uncertainty of triangular membership function.

In the following, the outline of the FAHP approach used in this study is discussed. If $X = \{x_1, x_2, \ldots, x_n\}$ is the object set, and the goal set is represented as $U = \{u_1, u_2, \ldots u_m\}$, each object is taken, and the extent analysis for each goal, $g_i$, is performed, respectively. Therefore, $m$ extent analysis values for each object can be obtained with the triangular fuzzy numbers, $M_{g,i}^j$, where $i = 1, 2, \ldots, n; j = 1, 2, \ldots, m$. $M_{g,i}^m$ represents the value of the extent analysis of the *ith* object for the *mth* goal.

For FAHP, this study defined the value of fuzzy synthetic extent $(S_i)$ as:

$$S_i = \sum_{j=1}^{m} M_{g_i}^j \otimes \left[ \sum_{i=1}^{n} \sum_{j=1}^{m} M_{g,i}^j \right]^{-1} \tag{7}$$

Perform the fuzzy addition operation $m$ extent analysis values for a particular matrix such that:

$$\sum_{i=1}^{n} \sum_{j=1}^{m} M_{g,i}^j = \left( \sum_{j=1}^{m} a_j, \sum_{j=1}^{m} b_j, \sum_{j=1}^{m} c_j \right), \ i = 1, 2, \ldots, n \tag{8}$$

Then, the inverse of the vector in Equation (8) is:

$$\left[ \sum_{i=1}^{n} \sum_{j=1}^{m} M_{g,i}^j \right]^{-1} = \left( 1/\sum_{i=1}^{n} c_i, 1/\sum_{i=1}^{n} b_i, \ 1/\sum_{i=1}^{n} a_i \right) \tag{9}$$

This study defined the value of $M_2$ $(V(M_2 \geq M_1))$, where $M_2 = (a_2, b_2, c_2) \geqslant M_1 = (a_1, b_1, c_1)$, and $d$ is the ordinate of the highest intersection point $D$ between $\mu_{M_1}$ and $\mu_{M_2}$ (Figure 3). $V(M_2 \geq M_1)$ is defined as:

$$V(M_2 \geq M_1) = \sup_{y \geq x} \left[ min \left( \mu_{M_1}(x), \mu_{M_2}(y) \right) \right] = hgt(M_1 \cap M_2)$$

$$= \mu_{M_2}(d) = \begin{cases} 1, & if\ b_2 \geq b_1 \\ 0, & if\ a_2 \geq c_1 \\ a_1 - c_2/(b_2 - c_2) - (b_1 - a_1), & otherwise \end{cases} \tag{10}$$

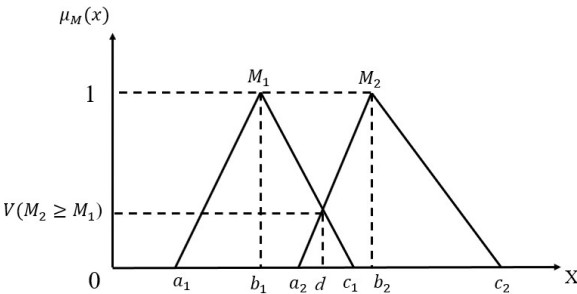

**Figure 3.** The value on the intersection between $M_1$ and $M_2$.

This study defined the value of $M$ ($V(M \geq M_1, M_2, \ldots, M_k)$), where $M_i(i = 1, 2, \ldots, k)$. $V(M \geq M_1, M_2, \ldots, M_k)$ can be defined by:

$$V(M \geq M_1, M_2, \ldots, M_k) = minV((M \geq M_i), i = 1, 2, 3, \ldots, k) \tag{11}$$

Assuming that $d(A_i) = min\ V(S_i \geq S_k)$, for $k = 1, 2, \ldots, n$; $k \neq i$, via normalization, the normalized weight vectors are $(d(A_1), d(A_2), \ldots, d(A_n))^T$. Ultimately, with FAHP's normalized weight vectors, six importance evaluations ($I_{im}$, $I_{vo}$, $I_{S_{PS}}$, $I_{S_I}$, $I_{S_M}$, $I_{S_L}$) were found (see Table 4).

**Table 4.** Evaluation ranking of MSE sustainability evaluation criteria for MSEs' $I_{im}$, $I_{vo}$, $I_{S_{PS}}$, $I_{S_I}$, $I_{S_M}$, $I_{S_L}$.

| Dimensions | $I_{im}$ | $I_{vo}$ | $I_{S_{PS}}$ | $I_{S_I}$ | $I_{S_M}$ | $I_{S_L}$ | Subdimensions | $I_{im}$ | $I_{vo}$ | $I_{S_{PS}}$ | $I_{S_I}$ | $I_{S_M}$ | $I_{S_L}$ | Criteria | $I_{im}$ | $I_{vo}$ | $I_{S_{PS}}$ | $I_{S_I}$ | $I_{S_M}$ | $I_{S_L}$ |
|---|---|---|---|---|---|---|---|---|---|---|---|---|---|---|---|---|---|---|---|---|
| GS | | | 3 | | | | GS$_1$ | 7 | 7 | 7 | 6 | 7 | 7 | GS$_{11}$ | 18 | 20 | 15 | 17 | 18 | 18 |
| | | | | | | | | | | | | | | GS$_{12}$ | 19 | 18 | 19 | 19 | 21 | 17 |
| | | | | | | | | | | | | | | GS$_{13}$ | 21 | 14 | 21 | 21 | 16 | 16 |
| | | | | | | | GS$_2$ | 4 | 4 | 5 | 4 | 5 | 5 | GS$_{21}$ | 8 | 9 | 11 | 11 | 10 | 11 |
| | | | | | | | | | | | | | | GS$_{22}$ | 6 | 7 | 5 | 9 | 6 | 7 |
| | | | | | | | | | | | | | | GS$_{23}$ | 15 | 21 | 22 | 12 | 20 | 19 |
| ES | | | 2 | | | | ES$_1$ | 5 | 5 | 4 | 5 | 4 | 2 | ES$_{11}$ | 13 | 15 | 7 | 16 | 9 | 6 |
| | | | | | | | | | | | | | | ES$_{12}$ | 10 | 13 | 9 | 14 | 8 | 3 |
| | | | | | | | | | | | | | | ES$_{13}$ | 16 | 16 | 17 | 18 | 13 | 15 |
| | | | | | | | ES$_2$ | 2 | 3 | 3 | 2 | 2 | 4 | ES$_{21}$ | 5 | 5 | 16 | 5 | 5 | 9 |
| | | | | | | | | | | | | | | ES$_{22}$ | 3 | 4 | 13 | 3 | 3 | 8 |
| | | | | | | | | | | | | | | ES$_{23}$ | 14 | 11 | 3 | 8 | 15 | 12 |
| FS | | | 1 | | | | FS$_1$ | 1 | 1 | 2 | 1 | 1 | 1 | FS$_{11}$ | 1 | 1 | 4 | 2 | 1 | 2 |
| | | | | | | | | | | | | | | FS$_{12}$ | 2 | 3 | 2 | 1 | 2 | 1 |
| | | | | | | | | | | | | | | FS$_{13}$ | 7 | 6 | 10 | 10 | 7 | 5 |
| | | | | | | | FS$_2$ | 3 | 2 | 1 | 3 | 3 | 3 | FS$_{21}$ | 4 | 2 | 6 | 4 | 4 | 4 |
| | | | | | | | | | | | | | | FS$_{22}$ | 9 | 8 | 8 | 6 | 12 | 13 |
| | | | | | | | | | | | | | | FS$_{23}$ | 11 | 10 | 1 | 7 | 11 | 14 |
| SS | | | 4 | | | | SS$_1$ | 6 | 6 | 6 | 8 | 8 | 6 | SS$_{11}$ | 12 | 12 | 12 | 13 | 17 | 10 |
| | | | | | | | | | | | | | | SS$_{12}$ | 17 | 22 | 14 | 23 | 24 | 23 |
| | | | | | | | | | | | | | | SS$_{13}$ | 20 | 23 | 18 | 24 | 23 | 24 |
| | | | | | | | SS$_2$ | 8 | 8 | 8 | 7 | 6 | 8 | SS$_{21}$ | 24 | 24 | 23 | 22 | 22 | 22 |
| | | | | | | | | | | | | | | SS$_{22}$ | 22 | 17 | 20 | 20 | 14 | 21 |
| | | | | | | | | | | | | | | SS$_{23}$ | 23 | 19 | 24 | 15 | 19 | 20 |

As shown in Table 4, which is the evaluation ranking of MSEs' sustainability evaluation criteria for MSEs' $I_{im}$, $I_{vo}$, $I_{S_{PS}}$, $I_{S_I}$, $I_{S_M}$, $I_{S_L}$. Table 5 shows the descriptive statistics of MSE sustainability evaluation criteria for MSEs' $I_{im}$, $I_{vo}$, $I_{S_{PS}}$, $I_{S_I}$, $I_{S_M}$, and $I_{S_L}$. As shown in Tables 4 and 5, the sequence of MSEs' important dimensions (from more to less) is the same for MSEs' perceived impact and environmental volatility from the COVID-19 pandemic and MSEs' perceived influence from SOIPE$_{PS}$, SOIPE$_I$, SOIPE$_M$, and SOIPE$_L$. This indicates that MSEs' major considerations in the different dimensions are the same, and that they would not differ based on an MSE's evaluation target. FS was found to be the most important dimension, with the importance of the other dimensions in decreasing order being ES, GS, and SS. This indicates that MSEs place a high value on FS, but care less about SS, which has been increasingly discussed in the literature.

**Table 5.** Descriptive statistics for $I_{im}$, $I_{vo}$, $I_{S_{PS}}$, $I_{S_I}$, $I_{S_M}$, $I_{S_L}$.

| | | $I_{im}$ | $I_{vo}$ | $I_{S_{PS}}$ | $I_{S_I}$ | $I_{S_M}$ | $I_{S_L}$ |
|---|---|---|---|---|---|---|---|
| | Mean | | | | 250 | | |
| | Median | 189.50 | 170.50 | 159.50 | 182.00 | 186.50 | 187.00 |
| Dimension | Min | 52 | 48 | 48 | 51 | 51 | 52 |
| | Max | 569 | 611 | 633 | 585 | 576 | 574 |
| | Std. Error | 228.67 | 252.22 | 263.56 | 237.53 | 233.93 | 232.73 |
| | Mean | | | | 125 | | |
| | Median | 85.50 | 63.50 | 93.00 | 63.50 | 85.00 | 95.00 |
| Subdimensions | Min | 12 | 18 | 13 | 25 | 18 | 19 |
| | Max | 410 | 416 | 392 | 390 | 448 | 469 |
| | Std. Error | 131.35 | 136.86 | 129.89 | 131.38 | 140.66 | 146.85 |
| | Mean | | | | 41.67 | | |
| | Median | 21.00 | 20.00 | 14.50 | 19.50 | 21.00 | 22.00 |
| Criteria | Min | 3 | 2 | 3 | 3 | 2 | 3 |
| | Max | 245 | 221 | 291 | 213 | 253 | 267 |
| | Std. Error | 53.93 | 55.56 | 62.47 | 50.60 | 56.42 | 57.95 |

As shown in Tables 4 and 5, the sequence of MSEs' important subdimensions (from more to less) is almost the same for MSEs' perceived impact and environmental volatility from the COVID-19 pandemic and MSEs' perceived influence from SOIPE$_{PS}$, SOIPE$_I$, SOIPE$_M$, and SOIPE$_L$. This indicates that MSEs' considerations in the different subdimensions are almost the same, but would vary slightly with a different MSE evaluation target in which the perceived cost–benefit for the MSE differs. Financial growth is the most important subdimension for most MSE evaluation targets, except for avoidance of financial risks for SOIPE$_{PS}$. The reason for this may be that MSEs consider financial growth the most, but the perceived cost of SOIPE$_{PS}$ is the most SOIPE; thus, MSEs care about the financial risks of SOIPE$_{PS}$.

From the subdimension results shown in Tables 4 and 5, MSEs' perceived impact and environmental volatility from the COVID-19 pandemic are MSEs' financial growth and avoidance of economic and financial risks, which are the same as MSEs' perceived influence from SOIPE$_{PS}$, SOIPE$_I$, and SOIPE$_M$. This indicates that MSEs consider economic and financial environmental volatility to be important, and SOIPE$_{PS}$, SOIPE$_I$, and SOIPE$_M$ might be able to help MSEs protect themselves from the impact and environmental volatility caused by the COVID-19 pandemic. Most MSEs' perceived influencing from SOIPE$_L$ are economic and financial growth and avoidance of financial risks. This indicates that MSEs care more about economic growth when they adopt SOIPE$_L$. As an increasing number

of MSEs in the industries of tourism, catering, hospitality, retail, transportation, culture, entertainment, sports, and real estate. All of these MSEs would like to adopt $SOIPE_L$, even closure or bankruptcy.

However, innovation governance organizations and avoidance of sociocultural and environmental risks should not comprise the impact and environmental volatility MSEs perceive most from the COVID-19 pandemic, which are the same as most MSEs' perceived influences from $SOIPE_{PS}$, $SOIPE_I$, $SOIPE_M$, and $SOIPE_L$. This indicates that $SOIPE_{PS}$, $SOIPE_I$, $SOIPE_M$, and $SOIPE_L$ should not be used to solve MSEs' problems related to governance organizations or sociocultural and environmental issues, for which MSEs did not show concern during the period of the COVID-19 pandemic.

As Tables 4 and 5 show, the order of importance MSEs place on these criteria (from more to less) is partly the same as that for MSEs' perceived impact and environmental volatility from the COVID-19 pandemic and perceived influences from $SOIPE_{PS}$, $SOIPE_I$, $SOIPE_M$, and $SOIPE_L$, which costs-benefits for MSE should not be the same. Thus, it is meaningful for this study to classify the influence of the COVID-19 pandemic on MSEs based on perceived impact and environmental volatility and classify SOIPE as $SOIPE_{PS}$, $SOIPE_I$, $SOIPE_M$, and $SOIPE_L$. Profitability and cost-down are the most important criteria for most MSEs' evaluation targets, but not for financial efficiency. This suggests that for MSEs, financial management is still focused on profits and costs, rather than financial efficiency.

As the criterion results in Tables 4 and 5 show, profitability, cost-down, cash flow, and interruption of the supply chain and demand chain are the top five criteria for MSEs' perceived impact and environmental volatility of the COVID-19 pandemic, which are the same as MSEs perceived influences from $SOIPE_I$ and $SOIPE_M$. This indicates that economic and financial environmental volatility are likely to be important for MSEs, and $SOIPE_I$ and $SOIPE_M$ may be able to precisely help MSEs protect themselves from the impact and environmental volatility of COVID-19. The top five criteria for MSEs' perceived influences from $SOIPE_{PS}$ are diversity, profitability, cost-down, industrial competitiveness and product and service quality, and budgets, taxes, and subsidies. This indicates that MSEs adopted $SOIPE_{PS}$ not only for cost benefits but also for diversity, competitiveness, taxes, and subsidies. The top five criteria for MSEs' perceived influences from $SOIPE_{PS}$ are profitability, cost-down, cash flow, financial efficiency, and short-term economic growth. This indicates that MSEs adopted $SOIPE_L$ not only for cost–benefit but also for cash flow, financial efficiency, and short-term economic growth.

### 4.3. Step 4: MSEs' Best SOIPE

As shown in Table 4 and Equations (1)–(5), Figures 4–6 present the evaluation of $Q_{S_{PS}}$, $Q_{S_I}$, $Q_{S_M}$, and $Q_{S_L}$ by the MSE sustainability evaluation dimensions and subdimensions. In Figures 4 and 5, $SOIPE_I$ is the best SOIPE based on the dimensions and subdimensions of MSE sustainability evaluation criteria. Regarding the reasons why $SOIPE_I$ is considered important, first, $SOIPE_I$ could reconnect an MSE's disrupted international business networks with innovation, learning, resources accessibility, international expansion, and marketing accessibility [47]. Second, $SOIPE_I$ could help MSEs access global customers, business partners, and demand and supply chain partners immediately, inexpensively, and reliably [48,49]. Third, $SOIPE_I$ could promote cross-border venture capital for MSEs, especially for those in emerging and less-developed economies [50–52]. Fourth, $SOIPE_I$ would become increasingly important as the COVID-19 pandemic persists, as the negative impact and MSEs' perceived environmental volatility of the pandemic could worsen, and the effects of $SOIPE_I$ noted above would be necessary for MSEs [50,53,54]. Fifth, $SOIPE_{PS}$, $SOIPE_M$, and $SOIPE_L$ should be promoted by $SOIPE_I$, as Hock-Doepgen et al. (2021) found that MSE knowledge management capabilities from $SOIPE_I$ lead to business model innovation of $SOIPE_{PS}$.

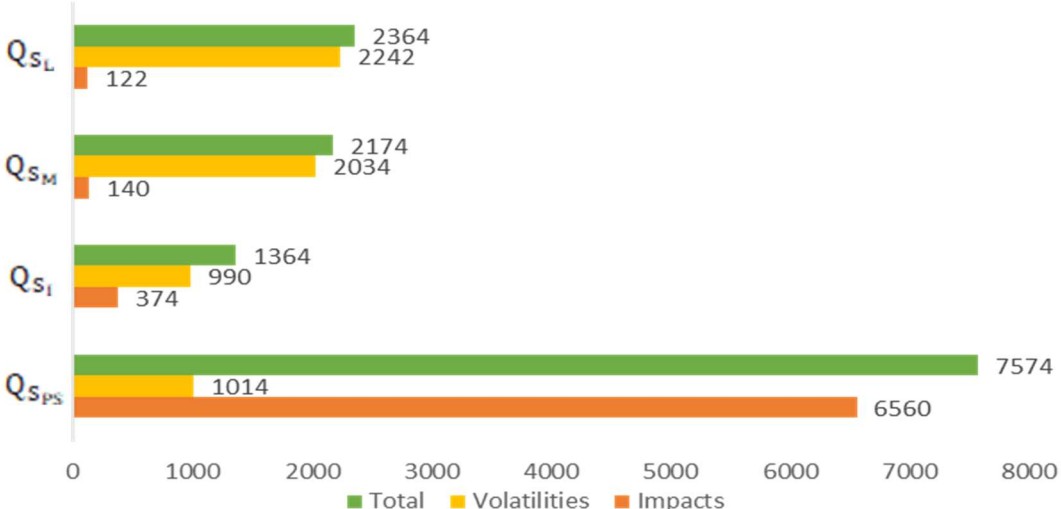

**Figure 4.** Evaluations of $Q_{S_{PS}}$, $Q_{S_I}$, $Q_{S_M}$, and $Q_{S_L}$ by MSE sustainability evaluation dimension.

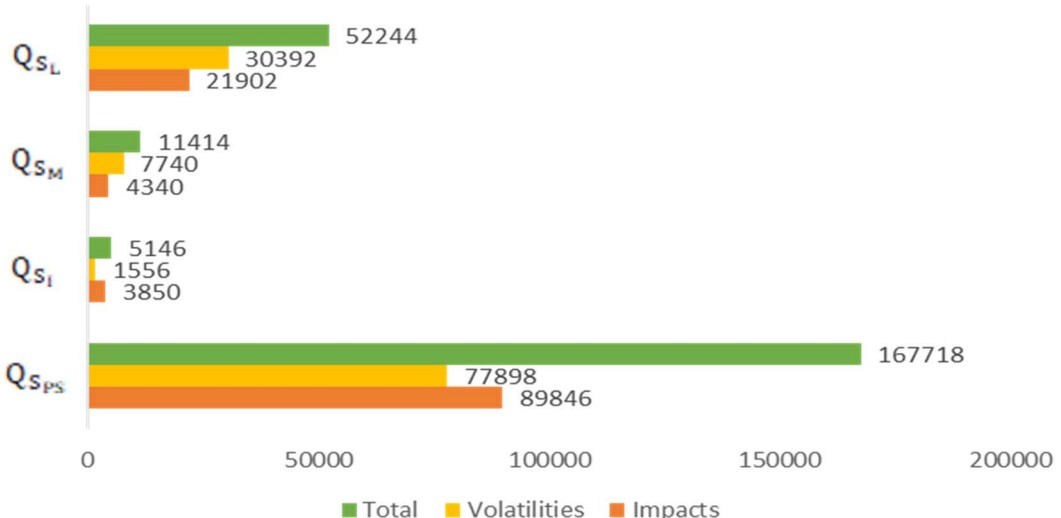

**Figure 5.** Evaluations of $Q_{S_{PS}}$, $Q_{S_I}$, $Q_{S_M}$, and $Q_{S_L}$ by MSE sustainability evaluation subdimension.

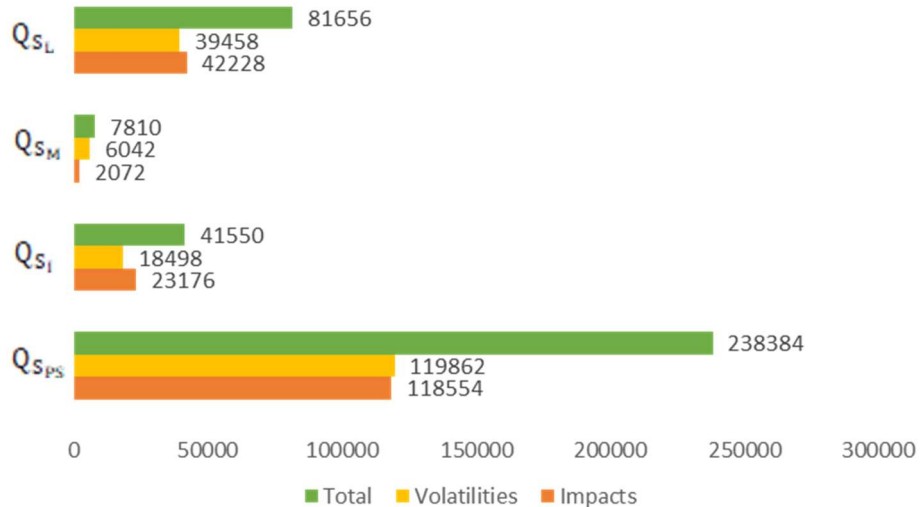

**Figure 6.** Evaluations of $Q_{S_{PS}}$, $Q_{S_I}$, $Q_{S_M}$, and $Q_{S_L}$ by MSE sustainability evaluation criteria.

Pratono [55] found this would not create a competitive advantage compared with the high information technological turbulence, organizational resilience, and marketing

communication that could result from that. This conclusion is in line with the results of the literature review. During the COVID-19 pandemic, several industries have grown, including enterprise technology services, home entertainment, artificial intelligence, robotics, telemedicine, e-commerce retailers, e-learning providers, courier pickup and delivery services, and cybersecurity. Further, the more MSEs there are, the cheaper and more effective coordination and productivity, and the more adaptable and speedy the responses will be to the perceived environmental volatility of the COVID-19 pandemic [10,56]. As recommended by the International Trade Centre [10], MSE's support organizations should gain access to timely, accurate, and credible information and solutions, collective actions for resilience, scale, and efficiency, early warning and risk reduction for global and local risks, and digital customer platforms for agility and competitiveness. As Gomes et al. [57] found, the innovation of small firms was mainly driven by external information sources, such as suppliers, trade fairs, and universities.

As Figure 6 shows, based on the results of MSE's sustainability evaluation model and procedure, SOIPE$_M$ is the best SOIPE for MSEs. This could explain the policies many countries and MSEs have used in response to the COVID-19 pandemic. SOIPE$_M$ might be able to help protect MSEs from the impacts and environmental volatilities caused by COVID-19 and has advantages such as geographical and temporal advantages, cost-saving, intermediator reduction, sales channels expansion, convenient management and communication, and competitiveness enhancement. Because SOIPE$_I$ might be unfamiliar to most MSEs, its results might be lower than its real evaluation. Thus, the present findings suggest that, to cope with the COVID-19 pandemic, MSEs could adopt SOIPE$_M$ and SOIPE$_I$.

Compared to the literature, the results in this paper proved SOIPE$_{PS}$ and SOIPE$_L$ might be good for MSEs under normal circumstances, but not during the COVID-19 pandemic; and logically proved why MSEs should adopt SOIPE$_M$ and SOIPE$_I$ to overcome the challenges of COVID-19. Previous research has expressed some doubts regarding SOIPE$_L$. First, some types of workers cannot be either substituted or allocated by SOIPE$_L$, as they are workers engaging in on-site COVID-19 prevention, such as doctors, nurses, policies, antiepidemic auxiliary workers, and transportation workers. Second, some MSEs would choose worker reduction and unpaid leave, but not SOIPE$_L$, especially in service industries such as education, public health, retail, hospitality, tourism, and construction. Finally, some SE employees may need reeducation or training to use SOIPE$_L$ [4,58,59].

*4.4. Discussions*

Considering the results of the comprehensive evaluation model and procedure for MSEs' sustainability evaluation and the case study of Chinese MSEs in the context of this study, existing literature often suggests that MSE innovation may positively influence performance, but less research has been performed in regard to the classification and verification of MSEs' SOIPE. As a contribution to the extant literature, we present a model to account for MSEs' selection mechanism under the causal relationship between MSEs' perceived negative impact and environmental volatility during a public emergency and MSE performance. MSEs' perceived negative impact and environmental volatility during a public emergency mediate the causal relationships between MSE performance and sustainability-oriented innovation, according to the study.

In addition, our research has several managerial implications for MSEs, including those in pro-emergency positions that can benefit from identifying the factors that influence MSE performance. First, MSE managers must justify the mechanism by which to select the optimal SOIPE for their companies. Findings from our study support the notion that emergency-impacted MSE performance depends on SOIPE and selection mechanisms. Second, an MSE's well-designed and effective SOIPE can help ensure their pro-emergency practices translate into improved performance, as they are built on accumulated emergency-impacted MSE capabilities. An MSE's optimal SOIPE can be aligned and balanced with pro-emergency practices in order to deliver better emergency-impacted MSE performance. Third, the mediating role of MSEs' SOIPE provides additional insights into the complex

mechanisms that underlie causal relationships between the negative impacts and perceived environmental volatility of a public emergency and MSE performance.

The limitations of this study are (1) Limited innovation modes: the innovation modes are not only the four mentioned in this paper but also include profit mode innovation, enterprise network innovation, enterprise structure (labor–capital ratio) innovation, enterprise process (process) innovation, enterprise system (product and service complementary efficiency) innovation, customer relationship, and brand innovation. (2) Single innovation mode: MSEs could choose one innovation mode and a combination of four innovation modes. (3) Limited samples: different sample structures might cause adjustment of empirical results. However, due to the length limitation of the article, it is not discussed, and it could be the topic of future research. (4) Method simplification: The evaluation content was not included in the discussion of quantitative indicators, and the results' robustness was not discussed.

## 5. Conclusions and Suggestions

During the public emergency created by the COVID-19 pandemic, MSEs have been observed to use SOIPE, including $SOIPE_{PS}$, $SOIPE_I$, $SOIPE_M$, and $SOIPE_L$, to defend against the pandemic's adverse effects. This study examined the selection criteria for MSEs' best SOIPE and used a case study of Chinese MSEs to evaluate it. Further, this study can provide suggestions for government and MSE innovation.

The study used a mixed-method design for the MSE sustainability evaluation model and procedure, set using the NGT, FAHP, and least-squares methods. Several relevant findings were observed. First, the adverse effects of the COVID-19 pandemic on MSEs could be thought of as MSEs' perceived impact and environmental volatility of the COVID-19 pandemic. Second, MSE SOIPE could be classified as $SOIPE_{PS}$, $SOIPE_I$, $SOIPE_M$, and $SOIPE_L$. Third, this study used a four-step comprehensive evaluation model and procedure for MSEs' sustainability, which comprises the initial list of MSE sustainability evaluation dimensions and criteria, final MSE sustainability evaluation criteria, FAHP questionnaire survey, and MSEs' best SOIPE. Fourth, the dimensions of the MSE sustainability evaluation criteria are as follows: GS, ES, FS, and SS.

Regarding the empirical results, first, MSEs tend to care about FS, but not SS. Second, MSEs' perceived impact and environmental volatility related to COVID-19 reflect their financial growth and avoidance of economic and financial risks. Third, MSEs' top five considerations related to the perceived impact and environmental volatility of the pandemic are profitability, cost-down, cash flow, and interruption of the supply chain and demand chain. Fourth, $SOIPE_M$ is the best for MSEs based on the sustainability evaluation criteria. Fifth, from the conclusions of Sections 3.2 and 3.3, this study's findings suggest that MSEs could adopt $SOIPE_M$ and $SOIPE_I$ to cope with the effects of the COVID-19 pandemic.

Future research could focus on the moderating effects of regional demographic characteristics on the causal relationships between MSEs' external environmental volatility during a public emergency and their performance. Orcos et al. [60] and Schwens et al. [61] showed that national institutions' variations of informal, formal, uncertainty avoidance, and market-supporting should be influencing factors on firms' cross-country variations. Future studies could also focus on the moderating effects of the institutional economy and market economy on the causal relationships and mediating effects identified in this study. Compared with the market economy, MSEs would be far more comfortable in an institutional economy with less MSE-perceived environmental volatility during a public emergency [62–64].

In addition, future studies could focus on a specific industry's characteristics to investigate the causal relationships and mediating effects discussed in this paper. Da Costa et al. [25] found that the automated and artisanal production processes used by Brazilian MSEs in the bakery industry are not significantly different for its competitiveness. This is because not only could $SOIPE_{PS}$ increase MSE production efficiency, but also MSEs' artisanal approaches can be appreciated by the public, which could be understood through $SOIPE_I$.

Future research could take a quantitative approach to explore the relationships among the variables of our study. Future research could also investigate how different emergency-related drivers can contribute to MSE performances. Future studies could focus on differences in the causal relationships between public emergencies and MSE performances and the mediating effects of SOIPE at different stages of the MSE lifecycle. Cao et al. [6] found that the causal relationships between MSEs' green innovation and their financial performances are related to the MSE development stage, which is significant in the initial and mature stages, but not in the growth stage.

**Author Contributions:** Conceptualization, W.H. and H.Y.; methodology, W.H. and S.C.; validation, H.Y. and S.C.; formal analysis, W.H. and H.Y.; investigation, H.Y.; resources, W.H. and H.Y.; data curation; writing—original draft preparation, H.Y. and S.C.; writing—review and editing, M.M. and S.C.; visualization, W.H. and M.M.; funding acquisition, W.H. All authors have read and agreed to the published version of the manuscript.

**Funding:** This research was funded by the Humanities and Social Sciences Projects of the Chinese Ministry of Education (Grant No. 21YJA790027).

**Institutional Review Board Statement:** Ethical review and approval were waived for this study because the survey with human subjects consisted of noninvasive items.

**Informed Consent Statement:** Informed consent was obtained from all the subjects involved in the study.

**Data Availability Statement:** The data presented in this research are not publicly available due to participant's privacy.

**Conflicts of Interest:** The authors declare no conflict of interest.

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
