# Peer review of "Micro- and Small-Sized Enterprises’ Sustainability-Oriented Innovation for COVID-19"

_sustainability, doi:10.3390/su14127521_

Round 1

Reviewer 1 Report

I learning more about small-sized enterprises in China. Some interesting results are shared, however methodology of the study is very confusing and not consistent throughout the paper.

 Introduction

·       -The overall aim of this paper is not clear.

·     -  I find the reference about Nigeria confusing and out of the scope of this study. Why the authors are presenting data only from Nigeria

·       -Explain and define GS,ES,FS and SS as well as sustainability oriented innovation

 Literature review

Some literature needs to be added. It might be good to split up the introduction in two: 1) introduction, identifying the research problem and the aim of the paper as well as background of the MSEs in China, and 2) Literature review – provide comprehensive review of the existing literature on the topics covered. I see some  review is placed under the methods but it should be moved earlier

Methodology

Methodology is confusing.

·      - The abstract said that this is a qualitative study, while later the paper talked about quantitative study and survey and at the end of the paper, the authors said that they are using mixed method approach. This needs to be revised.

·     -  Some parts of the methodology are actually a literature review. MSE background in China is too short. What are the sectors of the MSEs included in this study?

·     -  Refer to the Tables in the text

·    -   How did you choose your respondents for the interviews and the survey – sampling strategy?

·      - Table 3 is also confusing – there are 96 questionnaires collected but in frequency analysis there are only 52 responses?

Results/ discussion

It was difficult to follow the results. I am unclear where are results from the interviews.

Also not clear how the results of this study are compared to the literature

Conclusions

It should not be a summary of the paper but some concluding overall remarks derived from this study

Reviewer 2 Report

Title

Throughout the article it is found that authors relate this study with COVID-19, hence, I recommend to add COVID-19 in title too.

Abstract

Strating lines are very good, add your methodological techniques and study originality too

Introduction

Change the line “

 Sustainability should be an essential issue for modern enterprises, as measured by their corporate social responsibility (CSR) [1]”

In introduction show the emergent need of your research and introduce what you did and why you did? On the basis of research gap.

In simple words add a paragraph about the

Need of your research for academia and practitioners and present through state-of-art.

Methodology

This section is written well and need English improvement

Conclusion and suggestion

This section should be expended as

Implications for Theory

Implications for Practice

Limitations and Future Research Directions

Reviewer 3 Report

Dear authors,

1. The Introduction is too extensive, so you can synthesize it. The eliminated information can be reused in the Literature  Review section.

2. The research design can be improved through formulation of Hypotheses.

3. The Discussion section is missing. In this section it is recommended to describe the theoretical implications. Also you have to describe the limitation of your research.

Round 2

Reviewer 1 Report

Significant revision was made. Well done. I am still not clear why the authors need to provide details for the follow up quantitative study in this paper, which is based on qualitative data
